# Symptoms and signs of lung cancer prior to diagnosis: case–control study using electronic health records from ambulatory care within a large US-based tertiary care centre

Maria G Prado,[1] Larry G Kessler,[2] Margaret A Au,[1] Hannah A Burkhardt,[3] Monica Zigman Suchsland  ,[1] Lesleigh Kowalski,[1] Kari A Stephens,[1] Meliha Yetisgen,[3] Fiona M Walter,[4,5] Richard D Neal,[6] Kevin Lybarger,[7] Caroline A Thompson,[8,9] Morhaf Al Achkar  ,[1] Elizabeth A Sarma,[10] Grace Turner,[3] Farhood Farjah,[11] Matthew J Thompson  [1]

For numbered affiliations see end of article.

**Correspondence to**
Dr Matthew J Thompson;
mjt@uw.edu

## ABSTRACT

**Objective** Lung cancer is the most common cause of cancer-related death in the USA. While most patients are diagnosed following symptomatic presentation, no studies have compared symptoms and physical examination signs at or prior to diagnosis from electronic health records (EHRs) in the USA. We aimed to identify symptoms and signs in patients prior to diagnosis in EHR data.

**Design** Case–control study.

**Setting** Ambulatory care clinics at a large tertiary care academic health centre in the USA.

**Participants, outcomes** We studied 698 primary lung cancer cases in adults diagnosed between 1 January 2012 and 31 December 2019, and 6841 controls matched by age, sex, smoking status and type of clinic. Coded and free-text data from the EHR were extracted from 2 years prior to diagnosis date for cases and index date for controls. Univariate and multivariable conditional logistic regression were used to identify symptoms and signs associated with lung cancer at time of diagnosis, and 1, 3, 6 and 12 months before the diagnosis/index dates.

**Results** Eleven symptoms and signs recorded during the study period were associated with a significantly higher chance of being a lung cancer case in multivariable analyses. Of these, seven were significantly associated with lung cancer 6 months prior to diagnosis: haemoptysis (OR 3.2, 95% CI 1.9 to 5.3), cough (OR 3.1, 95% CI 2.4 to 4.0), chest crackles or wheeze (OR 3.1, 95% CI 2.3 to 4.1), bone pain (OR 2.7, 95% CI 2.1 to 3.6), back pain (OR 2.5, 95% CI 1.9 to 3.2), weight loss (OR 2.1, 95% CI 1.5 to 2.8) and fatigue (OR 1.6, 95% CI 1.3 to 2.1).

**Conclusions** Patients diagnosed with lung cancer appear to have symptoms and signs recorded in the EHR that distinguish them from similar matched patients in ambulatory care, often 6 months or more before diagnosis. These findings suggest opportunities to improve the diagnostic process for lung cancer.

## STRENGTHS AND LIMITATIONS OF THIS STUDY

⇒ Using natural language processing techniques to extract symptoms and signs from unstructured data provides a more complete dataset of clinical features presence compared with using coded data alone.

⇒ Case–control design recruited cases from ambulatory care population, and controls were randomly selected in a 10:1 ratio based on case clinic type, to reduce the possibility of bias.

⇒ Criteria for selection of cases and controls differed slightly; cases were selected based on a date of the first lung cancer diagnostic code in the electronic health record (EHR), whereas controls were selected based on having a visit to the matched type of clinic type within 3 months of the case diagnosis date.

⇒ Controls were not linked to cancer registry, so it is possible, though we believe highly unlikely, that there were a few cases among our controls who had a diagnosis of lung cancer in the cancer registry but no such diagnosis recorded in the EHR at any time (in our time window).

⇒ Availability and timing of symptom data for cases and controls is based on number and frequency of patient interactions with the healthcare system which could be due to a range of factors.

## INTRODUCTION

Lung cancer is the third most common cancer and the leading cause of cancer death in the USA.[1] Most patients with lung cancer are diagnosed following presentation to healthcare settings with symptoms or diagnosed incidentally, and many patients (47%) present with late-stage disease (stages 3 or 4).[2] Screening for lung cancer remains low in the USA, with an estimated 6.6% of adults

receiving screening in 2019.[3] [4] In addition to optimising screening, early detection efforts have focused on recognition of lung cancer symptoms with an overall goal of identifying patients at earlier, more treatable stages of the disease.[5–7] These symptoms range from 'alarm' symptoms, such as haemoptysis (a rare symptom), to relatively non-specific symptoms, such as persistent cough or unexpected weight loss.[6]

Diagnosing lung cancer based on non-specific symptom presentation is challenging, as these symptoms are more commonly associated with benign conditions or may be overlooked for long periods of time. A study of over 43 million patients using Medicare claims data identified a median time from symptom onset to diagnosis of approximately 6 months.[8] However, claims data lack the granularity needed to identify which clinical features patients present and how these might be used to differentiate patients with lung cancer from the vast majority of patients with benign conditions. To fill this gap, we examined the frequency and association of symptoms and physical examination signs in patients in ambulatory care prior to lung cancer diagnosis and matched controls.

## METHODS
### Study design
We performed a case–control study using data from the University of Washington Medicine (UWM) electronic health records (EHRs) and the Seattle/Puget Sound Surveillance, Epidemiology and End Results (SEER) Programme, a National Cancer Institute-supported national cancer registry.[9] A patient and caregiver stakeholder group was involved over a period of 2 years involving regular meetings in the design of this study and in the interpretation of the findings.

### Setting
Cases and controls were identified from patients who received ambulatory care at UWM, a large tertiary care academic health centre.

### Participants
Cases were identified from UWM patients aged 18 years or older, with a first primary lung cancer diagnosis (see International Classification of Diseases (ICD) 9 and 10 codes in online supplemental appendix 1) between 1 January 2012 and 31 December 2019, who had an established relationship with a UWM ambulatory care setting in the 2 years before the date of their first recorded lung cancer ICD code in the EHR (EHR diagnosis date). We chose the above study period because of the limited quality of the UWM EHR data prior to 2012. We defined ambulatory care as at least one encounter in family medicine, internal medicine, women's health, obstetrics and gynaecology, urgent care and/or emergency medicine. We used linkage to the regional SEER registry to verify cancer incident cases. Cases were excluded if they did not match with the SEER registry, or if they had a first primary

tumour located in anatomy other than the lung, or had evidence of a history of any of the following cancers identified using histology codes in SEER: tracheal cancer, mesothelioma, Kaposi sarcoma, lymphoma or leukaemia. Controls were identified from UWM patients with at least one encounter with the same type of ambulatory clinic within 3 months of the EHR diagnosis date of the index case (matching date). This 3-month window was chosen to avoid potential seasonal differences in respiratory symptoms. For each case, 10 controls were individually matched to the index case by age, sex (male, female), smoking status (ever vs never) and type of ambulatory care clinic where lung cancer case presented (emergency medicine vs other clinics listed above). We chose a 10:1 control: case match because we recognise the wide variety of patients presenting to ambulatory care settings. Controls were excluded if they had any lung cancer ICD codes in their EHR prior to their matched case diagnosis (index) date. Excluded cancers in cases (based on histology codes from the SEER registry) were not identified in controls as registry data were not available for controls. We also excluded any cases and controls who did not have any ICD codes in any encounter in the 2 years prior to diagnosis date (cases) or index date (controls) to ensure availability of data on prediagnosis symptoms and signs.

### Data collection
The UWM enterprise-wide data warehouse (EDW) was used to obtain data; this provides a central repository that integrates EHR across the UWM healthcare system including ambulatory care, specialty care and hospital services. Cases were identified during the study period using ICD codes (online supplemental appendix 1) and were linked to SEER to ensure accuracy of case identification and obtain history of previous cancers, histology (for exclusions and lung cancer type) and stage at diagnosis. The date of diagnosis was determined by date of pathology report at UWM. For cases that did not have a diagnosis through pathology or had a discrepancy greater than 30 days between date of pathology and first recorded lung cancer ICD code, two of three clinicians (MJT, LK and MAAc) reviewed the EHR of these cases to adjudicate dates. Controls were randomly sampled from within the matching strata, based on this adjudicated date of diagnosis.

Cases who had undergone lung cancer screening using low-dose CT within the 12 months prior to diagnosis date were identified from billing codes (Current Procedural Terminology or CPT 71271) and/or ICD codes (V76.0 (ICD-9) or Z12.2 (ICD-10).

An EHR data extraction protocol was applied to all encounters in the 2-year period prior and up to 6 months following the diagnosis date (cases) and index date (controls). These data composed of demographics (eg, age, sex, race, ethnicity), all ICD codes and CPT procedure codes linked to encounters such as laboratory tests, imaging procedures and pathology data. We also

extracted corresponding unstructured clinical notes for any of the above encounters from inpatient and outpatient settings. Clinical note types included progress notes, telephone encounters, hospital admission and discharge notes, notes of consultations with generalist and specialist clinicians, and nursing record notes. ICD codes recorded during the 2-year period prior to diagnosis for cases or prior to index date for controls were searched for the presence of 31 potential comorbidities to calculate the Elixhauser Comorbidity Index.[10] We excluded lung cancer ICD code information from this calculation. These index scores were then used to calculate van Walraven weighted scores for each patient, a range of −19 to 89.[11 12]

## Symptoms and signs

We identified symptoms and signs using coded data and unstructured data. A list of symptoms and signs which have previously been reported in cohort or case–control studies of individuals with lung cancer were identified from systematic reviews, hand review of individual studies and from contact with experts in oncology, cardiothoracic surgery and primary care (FMW, RDN, FF, MJT, see online supplemental appendix 2).[5 6 13–18] These were mapped to ICD codes, and used to search the extracted EHR coded data for any encounters that included any of these ICD codes in the 2-year observation period.

Symptoms and signs were automatically extracted from free-text clinical notes using natural language processing (NLP), including notes for all visit types in the 2-year period. In previous work, we developed a deep learning symptom extraction model that generates structured semantic representations of symptoms.[19] The annotation scheme and extraction architecture from this prior work represents symptoms using an event-based approach. Each symptom event includes a trigger span that identifies the specific symptom (eg, 'cough' or 'shortness of breath') and multiple attributes that characterise the symptom. The attributes most relevant to this work are the assertion value, which indicates whether the symptom is present, absent, possible, etc, and the anatomy, which indicates the anatomical location of the symptom (eg, 'chest wall' or 'lower back').

Structured symptom predictions were generated using the Span-based Event Extractor architecture in online supplemental appendix 3. Each clinical note is split into sentences, which feed into the extractor. The words (tokens) of each sentence are mapped to a vector space using a clinical version of the Bidirectional Encoder Representations from Transformers (BERT) model (no model fine-tuning). The BERT mapping of each sentence then feeds into a bidirectional Long Short-Term Memory network, which adapts the BERT encoding to the target extraction task. All possible token spans for the sentence are enumerated, and self-attention is used to create a representation for each span, $g_{c,i}$. Each of the enumerated spans is then classified using feedforward neural networks, $\phi_c$, that operate on the span representation, $g_{c,i}$. The span scoring layer, $\phi_c$, identifies the symptom triggers and attributes. Clinical notes frequently describe multiple symptoms within a sentence, and the relationships between the identified symptoms and attributes must be resolved. The identified symptom triggers are paired with the associated symptom attributes through the role scoring layer, $\psi_d$, which consists of a feedforward neural network that operates on span representation pairs. The output of the Span-based Event Extractor is a structured symptom representation, where identified symptoms are assigned multiple attributes.

In our original symptom work, we trained the Span-based Event Extractor on the COVID-19 Annotated Clinical Text Corpus (CACT).[19] To support the current research, we adapted the symptom extractor to the lung cancer domain. The domain adaptation involved creating the Lung Cancer Annotated Clinical Text (LACT) Corpus composed of 270 notes from patients with lung cancer (170 training and 100 test notes).[20] We trained the lung cancer symptom extractor by combining the CACT and LACT training sets. On the LACT test set, the lung cancer symptom extractor achieved 0.72 F1 for symptom identification and 0.65 F1 for assertion prediction. This extraction performance is comparable to the LACT inter-rater agreement of 0.82 F1 for symptom identification and 0.79 F1 for assertion prediction, indicating the model is achieving approximately human-level performance. We included the extracted symptoms and signs with assertion value present. All models were developed using the Python deep learning packages by PyTorch and Transformers.[21 22] The Span-based Event Extractor will be released through UW-BIoNLP github (https://github.com/uw-bionlp). The clinical notes will not be released for confidentiality purposes.

## Data analysis

Frequencies and counts were calculated for characteristics of cases and controls. The number of symptoms and signs obtained from coded data was compared with that obtained from free-text data using descriptive statistics. The proportion of patients with evidence of each symptom/sign occurring in the 2-year period prior to the diagnosis or index date was described for cases and controls. Odds of patients' case status, based on symptoms and signs identified from a combined dataset of coded and free-text data, were estimated using unadjusted conditional logistic regression. Symptoms and signs associated with lung cancer in unadjusted regressions (p<0.1) were included into multivariable conditional logistic regression analyses. We used the van Walraven comorbidity score to adjust for population differences in comorbidity burden. Analyses were repeated excluding symptom and sign data from 1, 3, 6 and 12 months before the diagnosis (or index) date. Lag times were chosen to provide information on the pattern of symptom-related visits over time and identify the symptoms and signs presenting furthest from diagnosis. We conducted secondary analyses investigating the potential effect of chronic respiratory disease (CRD) status, as defined by the presence of ICD codes

within the Elixhauser CRD subgroup, on the presence of symptoms and signs in the prediagnostic interval. We expected patients with CRD to present with symptoms and signs similar to those that present in early lung cancer. We assessed the effect of CRD by repeating the conditional logistic regression model including CRD as a covariate.

Statistical analyses were conducted using Python V.3.7 with the packages SciPy (V.1.4.1) and Statsmodels (V.0.11.1). The study was reported in line with the Strengthening the Reporting of Observational Studies in Epidemiology (STROBE) guidelines.[23]

### Patient and public involvement

We established a technical expert panel (TEP) that included patients with lung cancer and caregivers of patients with lung cancer. The TEP reflected on their personal experience with lung cancer symptoms as well as the lung cancer symptoms we identified in the EHR.

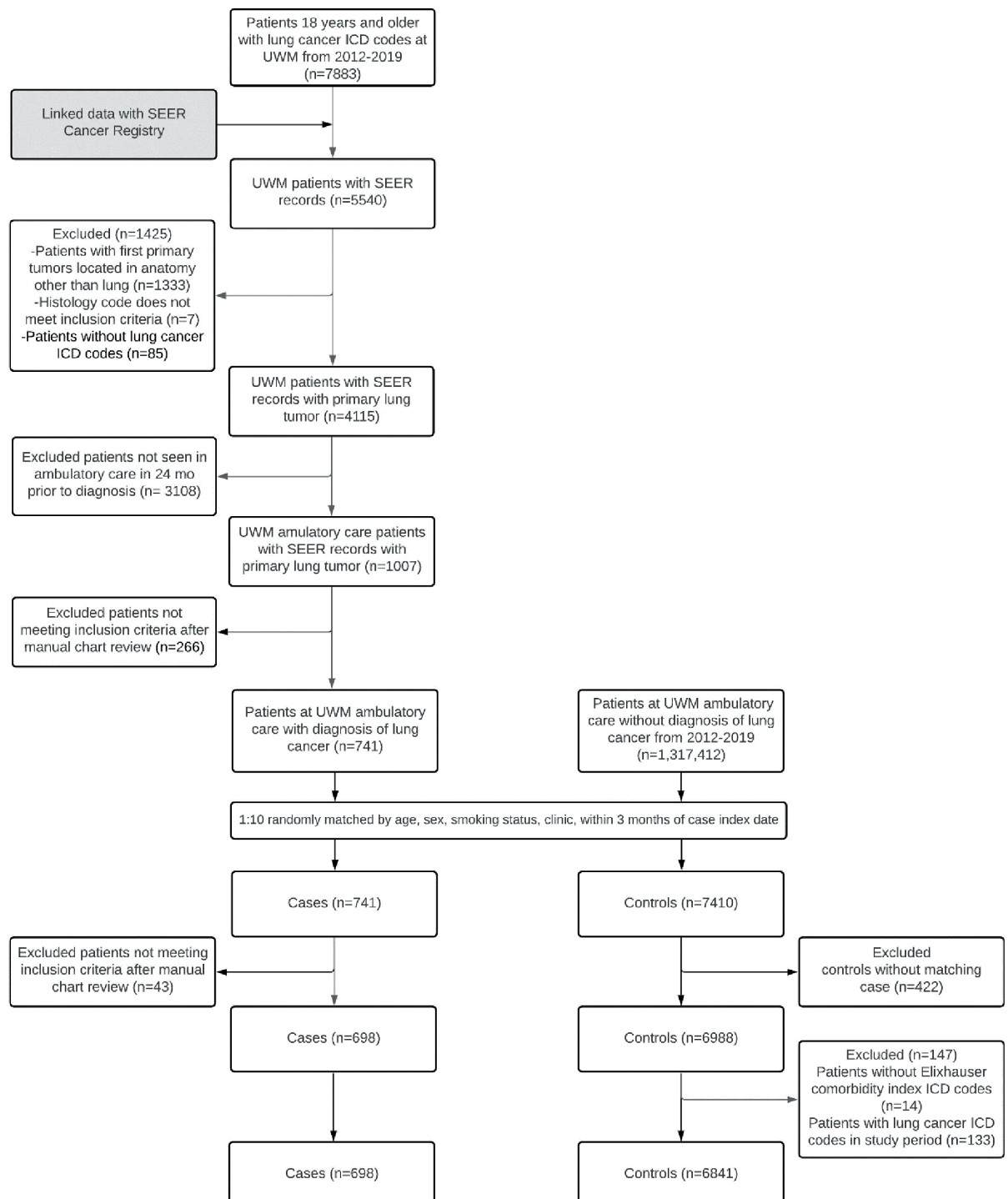

**Figure 1** Flow chart of case and control selection. ICD, International Classification of Diseases; SEER, Sound Surveillance, Epidemiology and End Results; UWM, University of Washington Medicine.

**Table 1** Characteristics of patients with lung cancer (cases) and matched controls in ambulatory care

| Characteristic | Cases (n=698) | Controls (n=6841) |
|---|---|---|
| Age, years | | |
| <60 | 161 (23.1%) | 1479 (21.6%) |
| 60–69 | 257 (36.8%) | 2514 (36.7%) |
| 70–79 | 183 (26.2%) | 1865 (27.3%) |
| 80+ | 97 (13.9%) | 983 (14.4%) |
| Race | | |
| American Indian or Alaska Native | 6 (0.9%) | 78 (1.1%) |
| Asian | 76 (10.9%) | 535 (7.8%) |
| Black or African American | 69 (9.9%) | 525 (7.7%) |
| Multiple races | 5 (0.7%) | 44 (0.6%) |
| Native Hawaiian or Other Pacific Islander | 4 (0.6%) | 40 (0.6%) |
| Unknown | 11 (1.6%) | 442 (6.5%) |
| White | 527 (75.5%) | 5177 (75.7%) |
| Ethnicity | | |
| Hispanic or Latino | 23 (3.3%) | 244 (3.6%) |
| Not Hispanic or Latino | 630 (90.3%) | 5782 (84.5%) |
| Unknown | 45 (6.4%) | 815 (11.9%) |
| Sex | | |
| Male | 353 (50.6%) | 3452 (50.5%) |
| Comorbidity—Elixhauser van Walraven Weighted Score, mean (SD) | 14.9 (11.6) | 4.4 (8.6) |
| No of clinic visits per patient, median (IQR) | | |
| In entire data window prior to diagnosis/index | 51.0 (28.0–97.8) | 23.0 (9.0–53.0) |
| In 1st quarter prior to diagnosis/index | 21.0 (12.0–35.0) | 5.0 (2.0–11.0) |
| In 2nd quarter prior to diagnosis/index | 7.0 (3.0–14.0) | 5.0 (2.0–11.0) |
| In 3rd quarter prior to diagnosis/index | 7.0 (3.0–12.0) | 5.0 (2.0–11.0) |
| In 4th quarter prior to diagnosis/index | 6.0 (3.0–13.0) | 5.0 (2.0–11.0) |

**Table 2** Comparison of frequency of symptoms and signs identified in coded or free-text data in cases compared with controls

| Symptom or sign | Cases (n=698) | Controls (n=6841) |
|---|---|---|
| Cough | 573 (82.1%) | 1654 (24.2%) |
| Shortness of breath | 515 (73.8%) | 1613 (23.6%) |
| Fatigue | 476 (68.2%) | 1587 (23.2%) |
| Ankle swelling | 447 (64.0%) | 1838 (26.9%) |
| Chest pain | 403 (57.7%) | 1401 (20.5%) |
| Chest crackles or wheeze | 397 (56.9%) | 575 (8.4%) |
| Back pain | 350 (50.1%) | 946 (13.8%) |
| Change in bowel habits | 336 (48.1%) | 1155 (16.9%) |
| Muscle weakness | 334 (47.9%) | 1102 (16.1%) |
| Fever | 322 (46.1%) | 1334 (19.5%) |
| Weight loss | 308 (44.1%) | 522 (7.6%) |
| Headache | 304 (43.6%) | 1205 (17.6%) |
| Dizziness | 299 (42.8%) | 1319 (19.3%) |
| Bone pain | 270 (38.7%) | 725 (10.6%) |
| Lack of appetite | 196 (28.1%) | 457 (6.7%) |
| Shoulder pain | 180 (25.8%) | 713 (10.4%) |
| Lymphadenopathy | 151 (21.6%) | 105 (1.5%) |
| Night sweats | 150 (21.5%) | 371 (5.4%) |
| Changes in sleep | 134 (19.2%) | 631 (9.2%) |
| Haemoptysis | 115 (16.5%) | 67 (1.0%) |
| Hoarseness | 67 (9.6%) | 133 (1.9%) |
| Finger clubbing | 39 (5.6%) | 2 (0.0%) |

They discussed and advised on study methods, data analysis, and communication and visualisation of results.

## RESULTS
### Participants
#### Selection of cases and controls
A total of 7883 patients with lung cancer ICD codes were identified in the UWM EDW over the study period. Following linkage of these patients and those identified as having a primary lung tumour from SEER, 4115 patients were identified common to both, including 741 cases. After matching 7410 controls, a chart review resulted in exclusion of 43 additional cases. Controls that were matched to these 43 cases were excluded (n=422), resulting in 698 cases matched to 6841 controls (figure 1).

#### Description of cases and controls
Cases and controls were similar in terms of sex and race (cases 50.6% male, 75.5% white; controls 50.5% male, 75.7% white, see table 1), as well as ethnicity (cases 3.3% Hispanic, controls 3.6%). Cases had higher comorbidity scores (M=14.9, SD=11.6) than controls (M=4.4, SD=8.6). Cases also had a greater median number of healthcare visits over the 2-year period prior to diagnosis (51.0, 95% CI 28.0 to 97.8) than controls (23.0, 95% CI 9.0 to 53.0). The difference in median number of healthcare visits was greater in the last 3-month period prior to the diagnosis/index date (cases 21.0, 95% CI 12.0 to 35.0 vs controls 5.0, 95% CI 2.0 to 11.0) than in the 2nd, 3rd or 4th quarters prior to diagnosis. The stage distribution of cases was as follows: stage 1%–29%, stage 2%–7%, stage 3%–17% and stage 4%–42% (5% were stage 0 or unknown stage).

### Frequency of symptoms and signs extracted from coded and free-text data
Of the 22 symptoms and signs that we systematically examined, NLP identified 20 of the 22 symptoms and signs in greater proportions of patients affected than from the coded data alone (see online supplemental appendix 4). In comparison to coded data, we saw a range of 12.9%–97.6% greater symptom and signs reports with NLP of textual clinical notes. In contrast, a greater proportion of patients had two symptoms and signs (shoulder pain, lymphadenopathy) identified from coded rather than free-text data.

Table 3  Univariate and multivariate analyses of symptoms and signs identified in coded or free-text data of cases compared with controls, adjusted for comorbidity (descending order by multivariate ORs)

| Symptom or sign | Univariate OR (95% CI) | Multivariate OR (95% CI) | Multivariate P value |
|---|---|---|---|
| Finger clubbing | 175.7 (40.1 to 770.0)* | 50.1 (8.9 to 283.3) | <0.0001 |
| Lymphadenopathy | 9.4 (6.9 to 12.8)* | 5.8 (3.8 to 8.8) | <0.0001 |
| Cough | 11.1 (8.8 to 13.9)* | 4.7 (3.5 to 6.3) | <0.0001 |
| Haemoptysis | 14.5 (10.2 to 20.8)* | 3.5 (2.2 to 5.5) | <0.0001 |
| Chest crackles or wheeze | 9.9 (8.1 to 12.2)* | 3.2 (2.4 to 4.3) | <0.0001 |
| Weight loss | 5.9 (4.8 to 7.2)* | 2.9 (2.2 to 3.9) | <0.0001 |
| Back pain | 4.7 (3.9 to 5.7)* | 2.4 (1.8 to 3.1) | <0.0001 |
| Bone pain | 4.6 (3.8 to 5.7)* | 2.3 (1.7 to 3.1) | <0.0001 |
| Shortness of breath | 6.0 (4.9 to 7.3)* | 1.9 (1.4 to 2.5) | <0.0001 |
| Fatigue | 4.8 (4.0 to 5.8)* | 1.8 (1.4 to 2.4) | <0.0001 |
| Chest pain | 3.6 (3.0 to 4.3)* | 1.4 (1.1 to 1.8) | 0.0118 |
| Shoulder pain | 2.3 (1.8 to 2.8)* | 1.3 (1.0 to 1.7) | 0.1111 |
| Ankle swelling | 3.3 (2.7 to 4.0)* | 1.1 (0.9 to 1.5) | 0.3643 |
| Headache | 2.5 (2.1 to 3.0)* | 1.1 (0.8 to 1.4) | 0.5619 |
| Hoarseness | 3.5 (2.5 to 5.0)* | 1.1 (0.7 to 1.7) | 0.8447 |
| Change in bowel habits | 3.0 (2.5 to 3.6)* | 1.0 (0.8 to 1.4) | 0.8880 |
| Muscle weakness | 2.9 (2.4 to 3.5)* | 1.0 (0.7 to 1.3) | 0.9581 |
| Night sweats | 3.3 (2.6 to 4.2)* | 0.8 (0.6 to 1.2) | 0.2998 |
| Lack of appetite | 2.6 (2.1 to 3.3)* | 0.7 (0.5 to 0.9) | 0.0193 |
| Dizziness | 2.0 (1.7 to 2.4)* | 0.6 (0.4 to 0.8) | 0.0004 |
| Changes in sleep | 1.3 (1.1 to 1.7)* | 0.5 (0.3 to 0.6) | <0.0001 |
| Fever | 2.1 (1.7 to 2.5)* | 0.4 (0.3 to 0.6) | <0.0001 |

Conditional logistic regression models adjusted for comorbidities using van Walraven weighted score with each symptom or sign modelled individually (univariate) and mutually adjusted (multivariate).
*Significant at p<0.0001 for univariate analysis.

## Comparison of frequency of symptoms and signs between cases and controls

The frequency of all 22 symptoms and signs examined was higher in cases than controls (see table 2). Moreover, the ranking of symptoms and signs differed slightly between cases and controls, with cases reporting cough (82.1%), shortness of breath (73.8%), fatigue (68.2%), ankle swelling (64.0%) and chest pain (57.7%), whereas controls reported ankle swelling (26.9%), cough (24.2%), shortness of breath (23.6%), fatigue (23.2%) and chest pain (20.5%) most frequently. Haemoptysis occurred relatively infrequently among cases (16.5%) and rarely among controls (1.0%).

## Univariate associations of symptoms and signs between cases and controls

In models adjusted for comorbidity score, when considered independently, all 22 symptoms and signs had ORs that were significantly different between cases and controls (all p<0.0001, see table 3). The symptoms and signs with the largest OR significantly associated with a higher chance of being a case were finger clubbing (OR 175.7, 95% CI 40.1 to 770.0), haemoptysis (OR 14.5, 95% CI 10.2 to 20.8), cough (OR 11.1, 95% CI 8.8 to 13.9), chest crackles or wheeze (OR 9.9, 95% CI 8.1 to 12.2) and lymphadenopathy (OR 9.4, 95% CI 6.9 to 12.8).

## Multivariable associations of symptoms and signs between cases and controls

We included all 22 symptoms and signs from the univariate analysis and comorbidity score in a multivariable analysis. After mutual adjustment, 15 had significant ORs (all p<0.05, see table 3). The presence of 11 symptoms and signs were associated with a significantly higher odds of being a case, with ORs ranging from 1.4 (chest pain) to 50.1 (finger clubbing). The largest ORs were noted for finger clubbing (OR 50.1, 95% CI 8.9 to 283.3), lymphadenopathy (OR 5.8, 95% CI 3.8 to 8.8), cough (OR 4.7, 95% CI 3.5 to 6.3), haemoptysis (OR 3.5, 95% CI 2.2 to 5.5) and chest crackles or wheeze (OR 3.2, 95% CI 2.4 to 4.3). In contrast, the presence of four symptoms was associated with a significantly higher odds of being a control: fever (OR 0.4, 95% CI 0.3 to 0.6), changes in sleep (OR 0.5, 95% CI 0.3 to 0.6), dizziness (OR 0.6, 95% CI 0.4 to 0.8) and lack of appetite (OR 0.7, 95% CI 0.5 to 0.9).

We repeated the multivariable analysis, excluding symptoms and signs recorded in periods of 1, 3, 6 and 12 months prior to diagnosis (see figure 2). Some symptoms

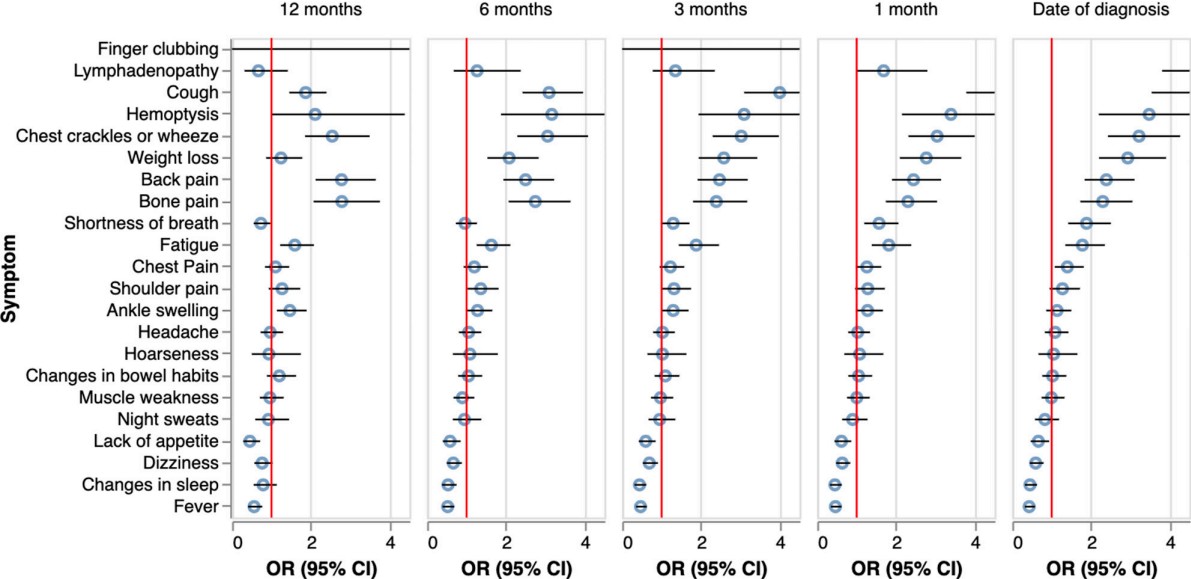

**Figure 2** Multivariable analysis of symptoms or signs of cases compared with controls with symptom and sign data excluded from 1, 3, 6 and 12 months prior to diagnosis/index date. Mutual adjustment of all symptoms and signs in using a conditional logistic regression model stratified by time prior to date of diagnosis. Models additionally adjusted for comorbidities using van Walraven weighted score. For the complete set of results, see online supplemental appendix 5.

and signs remained significantly associated with cases up to 6 months prior to diagnosis (cough, haemoptysis, chest crackles and wheeze, weight loss, back pain, bone pain, fatigue). Of these, all except weight loss were also significantly associated with cases 12 months prior to diagnosis. Other symptoms and signs became significantly associated with being a case closer to the date of diagnosis: shortness of breath and chest pain (3 months prior to diagnosis), lymphadenopathy and finger clubbing (1 month prior) (see online supplemental appendix 5).

## Secondary analyses

To determine whether the associations were robust to the presence of CRD, we performed a secondary conditional logistic regression that was adjusted for CRD, along with all our matching variables and comorbidity score. The presence of CRD appeared to have no statistically significant effect when directly added as a covariate (OR 1.05, 95% CI 0.81 to 1.36, p=0.7229, see online supplemental appendices 6,7).

## DISCUSSION
## Main findings

This is the first case–control study in the USA to use routine, prospectively collected EHR data to describe the frequency of symptoms and signs of lung cancer and estimate associations with incident lung cancer cases compared with non-lung cancer patients receiving routine ambulatory care in the same time period. Our findings provide unique information on symptoms and signs associated with a higher chance of a patient in ambulatory care being diagnosed with lung cancer, and the duration of these associations prior to their cancer diagnosis. In contrast to prior work on national databases,

extracting clinicians' documentation of clinical features from their free-text clinical notes using NLP provided more complete symptom identification data, rather than relying on data available only in coded, structured data collected in routine care. Our findings provide evidence-based, quantitative support for the development of decision rules around the diagnostic workup of symptomatic patients, which could lead to the improvement of earlier diagnosis of lung cancer. Of the 22 symptoms and signs studied, 11 were found in adjusted models to be associated with a higher chance of being a lung cancer case, and most of these 11 were present and still significantly associated up to 12 months prior to diagnosis; this suggests opportunities for improved screening practices that may lead to earlier diagnosis and possibly improved outcomes.

Our findings also suggest that the clinical presentation of lung cancer appears to be similar, regardless of the presence of other comorbidities, CRD or smoking. For patients and clinicians, this is important as several of the symptoms or signs we identified may currently be dismissed as being attributable to underlying smoking or comorbid conditions.

## Comparison with existing literature

Several of the symptoms and signs we found as having statistically significant ORs have been identified in studies using data from ambulatory care in other healthcare systems, especially haemoptysis and cough. However, among the symptoms and signs Hamilton *et al* found to be associated with being a lung cancer case in the UK, loss of appetite had the highest OR (86.0), whereas we failed to identify an association with lung cancer.[5] This may be due to a difference in study populations or our use of NLP in EHR data.

Our findings also provide evidence of the temporality of a 'clinical signal' for lung cancer based on symptoms and signs documented in the EHR, at least 6 and up to 12 months prior to diagnosis, consistent with a Medicare claims study. Data from our study and Nadpara *et al* study, which used claims data, provide evidence for time intervals from first presentation with symptoms to diagnosis that are on the upper range (6 months) of those reported using analysis of coded symptoms in primary care databases in several UK and European studies.[8] These describe the overall time interval from first symptom recording in medical records to diagnosis ranging from 3 to 6 months.[6 24 25] While not directly comparable, qualitative research from patients with lung cancer and caregivers describe changes noticeable to the individual more than 12 months before attending a healthcare visit.[17 26 27]

## Strengths and limitations
Using NLP to extract symptoms and signs from unstructured data allowed us to capture a more complete dataset of symptom presence compared with using coded data alone. We selected cases from an empaneled ambulatory care population, where we expected EHR data would be available for the period of interest in this study and attempted to exclude patients who were attending only for secondary or tertiary care provided at UWM. Controls were randomly selected based on case clinic type, to reduce the possibility of bias, and duration of follow-up time and availability of data for cases and controls were similar, particularly in visit frequency. We used a robust design where we matched 10 controls to 1 case, providing greater power and precision, and matched on smoking so that our analyses could not be confounded based on ever versus never exposure to smoking.

Limitations included criteria for selection of cases and controls differed slightly. As is customary in incident case–control studies, cases were selected based on a diagnosis date defined as the date of the first lung cancer ICD code in the EHR. In this way, we captured the diagnostic path from symptom presentation to diagnosis for all cases. Controls were selected based on having a visit to the matched case clinic type (to account for difference in emergency vs other forms of ambulatory care) within 3 months of the case diagnosis date, however, the timing of control selection does not necessarily reflect a 'pathway to diagnosis' for some other condition, just recent routine care. Additionally, because we did not link to SEER for the control population, we were unable to apply two of the case exclusion criteria to our control sample: (1) no current or prior history of lung cancer in SEER, although we did check the UW EHR for concurrent lung cancer-related ICD codes and medical history so this should be rare and (2) no prior history of tracheal cancer, mesothelioma, Kaposi sarcoma, lymphoma or leukaemia in SEER. Additionally, EHR data can sometimes be subject to misclassification. For example, detailed EHR smoking history may be unreliable and the EHR does not reliably capture health literacy or socioeconomic status; however, we used a very broad definition of smoking (ever vs never) and used a comorbidity score to control for health status. Finally, availability and timing of symptom data for cases and controls is based on patient interactions with the healthcare system, not a prespecified protocol of data collection. Patients who have more contact with their providers (which could be due to a range of factors) may have had more data captured.

## Implications for clinicians, researchers policy-makers
Differentiating patients who may have symptoms or signs of lung cancer from those attending ambulatory care is a critical and challenging step in the earlier detection of this cancer. Our findings not only identify the 'red flag' (highly specific, but infrequent) symptoms and signs that primary care providers should be aware of (eg, haemoptysis), but also highlight which of a larger range of 'non-specific' symptoms and signs should equally raise suspicion such as bone pain and weight loss. Furthermore, our findings support the importance of clinical documentation, and continuity of care to identify and act on sustained changes in patients' clinical presentations.

Confirmation of our findings using datasets from other healthcare systems in the USA is needed and could be enhanced by more advanced machine learning modelling to incorporate additional clinical variables including quantitative data such as changes in body weight or results of routinely collected laboratory tests, given emerging evidence for associations between weight loss and minor deviations of haemoglobin or platelet count with incident cancer.[28] Given the low uptake of low dose CT screening for lung cancer in the USA, our findings provide support for revising current priorities to improve early diagnosis of lung cancer.[29]

## CONCLUSIONS
Patients in ambulatory care settings who are subsequently diagnosed with lung cancer appear to have symptoms and signs that distinguish them from other patients, often months before lung cancer diagnosis. To improve earlier detection of lung cancer, interventions are urgently needed that promote earlier screening based on symptomatic presentations in ambulatory care that may lead to an earlier detection and treatment of lung cancer.

**Author affiliations**
[1]Department of Family Medicine, University of Washington, Seattle, Washington, USA
[2]Health Services, University of Washington, Seattle, Washington, USA
[3]Department of Biomedical Informatics and Medical Education, University of Washington, Seattle, Washington, USA
[4]Wolfson Institute of Population Health, Barts and The London School of Medicine and Dentistry, Queen Mary University of London, London, UK
[5]The Primary Care Unit Department of Public Health and Primary Care, University of Cambridge, Cambridge, UK
[6]University of Exeter, Exeter, UK
[7]Department of Information Sciences and Technology, George Mason University, Fairfax, Virginia, USA

[8]Department of Epidemiology, The University of North Carolina, Chapel Hill, North Carolina, USA
[9]Division of Epidemiology and Biostatistics, San Diego State University, San Diego, California, USA
[10]National Cancer Institute, NIH, Bethesda, Maryland, USA
[11]Department of Surgery, University of Washington, Seattle, Washington, USA

Acknowledgements We would like to thank the patients and clinicians at University of Washington Medicine, and the members of our TEP.

Contributors MGP extracted data from UW Medicine and linked to SEER Cancer Registry, supported study management and execution, wrote the manuscript, provided critical comments, edited the manuscript and approved its final version. LGK assisted with design of the study and supported its execution, provided advice and expertise for study design, analyses and interpretation of data, wrote the manuscript, provided critical comments, edited the manuscript and approved its final version. MAAu performed the analyses, provided advice and expertise for study design, conducted analyses and interpretation of data, provided critical comments, edited the manuscript and approved its final version. HAB supported data extraction and data linkage, assisted with analyses, created figures and tables, assisted with interpretation of data, provided critical comments, edited the manuscript and approved its final version. MZS assisted with design of the study and supported its execution, extracted data from UW Medicine and linked to SEER Cancer Registry, provided further advice and expertise for study design, and interpretation of data, provided critical comments, edited the manuscript, and approved its final version. LK assisted with design of the study and supported its execution, provided advice and expertise for study design, clinical interpretation of data, provided critical comments, edited the manuscript, and approved its final version. KAS assisted with design of the study, extracted data from UW Medicine and linked to SEER Cancer Registry, provided advice and expertise for study design, interpretation of data, provided critical comments, edited the manuscript and approved its final version. MY created the natural language annotation tool and extracted free-text data, assisted with interpretation of data, provided critical comments, edited the manuscript, and approved its final version. FMW provided advice and expertise for study design, clinical input and interpretation of data, provided critical comments, edited the manuscript and approved its final version. RDN provided advice and expertise for study design, clinical input and interpretation of data, provided critical comments, edited the manuscript, and approved its final version. KL created the natural language annotation tool and extracted free-text data, assisted with interpretation of data, provided critical comments, edited the manuscript and approved its final version. CAT provided advice and expertise for study design, analytical methods and interpretation of data, provided critical comments, edited the manuscript and approved its final version. MAAc provided advice and expertise for study design, clinical input and interpretation of data, provided critical comments, edited the manuscript, and approved its final version. EAS provided advice and expertise for study design, clinical input and interpretation of data, provided critical comments, edited the manuscript, and approved its final version. GT supported implementation of the natural language annotation tool and extracted free-text data, assisted with interpretation of data, provided critical comments, edited the manuscript, and approved its final version. FF provided advice and expertise for study design, clinical input and interpretation of data, provided critical comments, and approved its final version. MJT was the principal investigator for the study and is its guarantor, designed the study and supervised its execution, provided clinical guidance, interpreted data, wrote the manuscript, edited the manuscript, and approved its final version.

Funding This research was funded by the Gordon and Betty Moore Foundation through Grant GBMF8837 to the University of Washington. This research is linked to the CanTest Collaborative, which is funded by Cancer Research UK (C8640/A23385), of which FMW is Director and RDN and MJT are Associate Directors. This research was supported by the Cancer Surveillance System of the Fred Hutchinson Cancer Research Center, which is funded by Contract Numbers HHSN261201800004I and N01 PC-2018-00004 from the Surveillance, Epidemiology and End Results (SEER) Program of the National Cancer Institute with additional support from the Fred Hutchinson Cancer Research Center and the State of Washington.

Disclaimer The views expressed are those of the authors and do not necessarily represent the official position of the National Cancer Institute, the National Institute of Health, or Department of Health and Human Services.

Competing interests None declared.

Patient and public involvement Patients and/or the public were involved in the design, or conduct, or reporting, or dissemination plans of this research. Refer to the Methods section for further details.

Patient consent for publication Not applicable.

Ethics approval The study was conducted according to the guidelines of the Declaration of Helsinki and was approved by the University of Washington Human Subjects Division (STUDY 000013191). Consent is not required to use electronic medical record data which has been anonymised and used for research purposes, this was approved by the institution IRB.

Provenance and peer review Not commissioned; externally peer reviewed.

Data availability statement Data are available on reasonable request. Fully anonymised data may be available on reasonable request to the corresponding author, once appropriate data sharing and ethics approvals have been obtained.

ORCID iDs
Monica Zigman Suchsland http://orcid.org/0000-0001-7007-6973
Morhaf Al Achkar http://orcid.org/0000-0002-4160-0550
Matthew J Thompson http://orcid.org/0000-0003-0256-8444

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
