## [Reviewer comments · BMJ Open]

ARTICLE DETAILS

TITLE (PROVISIONAL)	Symptoms and signs of lung cancer prior to diagnosis: Case-control study using electronic health records from ambulatory care within a large US-based tertiary care center
AUTHORS	Prado, Maria G.; Kessler, Larry; Au, Margaret A; Burkhardt, Hannah; Zigman Suchsland, Monica; Kowalski, Lesleigh; Stephens, KA; Yetisgen, Meliha; Walter, Fiona M; Neal, Richard; Lybarger, Kevin; Thompson, Caroline; Al Achkar, Morhaf; Sarma, Elizabeth; Turner, Grace; Farjah, Farhood; Thompson, Matthew

VERSION 1 – REVIEW

REVIEWER	Ruth Swann Cancer Research UK
REVIEW RETURNED	18-Nov-2022

GENERAL COMMENTS	Thank you for the opportunity to review this paper. The authors address an important question and explain and present this well. I have made some comments as shown below. Strengths and limitations of the study As highlighted in the discussion, an additional limitation of this study is that the controls were not linked to SEER. Introduction 'Screening for lung cancer remains low in the US.' For the non-US audience, it would be useful to know the approximate uptake of the lung cancer screening programme in the US. Methods As the data are collected from activity at UWM, a large tertiary care health centre, is it possible that cases or controls could have had consultations related to lung cancer in primary care or other hospitals before they were admitted to UWM? Would the SEER diagnosis date be more accurate for the date of lung cancer diagnosis rather than the EHR diagnosis date? To explain why the 3 month window was chosen, the following text in the discussion would be more useful in the methods '(to avoid potential seasonal differences in respiratory symptoms)'. ' Could the authors briefly explain why cases with lymphoma, leukaemia, Kaposi sarcoma were excluded? The methods describe that cases were excluded if they had evidence of previous tracheal cancer, mesothelioma, Kaposi
---

	sarcoma, lymphoma, or leukemia. Was evidence of these conditions searched for in the controls? Why was race not included when matching controls to cases? Was the analysis multivariate or multivariable? Both terms have been used in the paper. Discussion Regarding the text 'Additionally, because we did not link to SEER for the control population, we were unable to apply two of the case exclusion criteria to our control sample: no current or prior history of lung cancer in SEER, although we did check the UW EHR for concurrent lung-cancer related ICD codes and medical history so this should be rare, and no prior history of tracheal cancer, mesothelioma, Kaposi sarcoma, lymphoma, or leukemia in SEER.' This is slightly confusing as the first part of the paragraph says that controls were not linked to SEER, then the end of the paragraph says that the controls had no prior history of the listed cancer types in SEER. Could it be possible that a control had a lung cancer diagnosis after the index date? As some of these patients also have lung cancer related symptoms, could it be possible that they are just on an earlier part of the lung cancer diagnosis pathway? Table 1 Why are the Hispanic or Latino group displayed separately in the ethnicity category? There is no reference to this in the text. Figure 1 Here it mentions that patients with first primary tumour located in anatomy other than lung are excluded, in the methods it mentions that cases were excluded if they had evidence of a history of any of the following cancers identified using histology codes in SEER: tracheal cancer, mesothelioma, Kaposi sarcoma, lymphoma, or leukemia (which I presume is histology code does not meet inclusion criteria in figure 1). Could you please confirm whether other primary tumours were excluded from the cases and if so include this in the methods. Figure 2 As not all of the results are shown in the graph, I would suggest to add in the footnote to see Appendix 4 for complete set of results.
--	--

REVIEWER	Alejandro Rodríguez-González Universidad Politecnica de Madrid
REVIEW RETURNED	26-Nov-2022

GENERAL COMMENTS	The paper is quite interesting and well-written. The methods are, in general, well described, as well as the results obtained. My main concern is with the part where the NLP approach is described. The description of the NLP approach for the identification of potential symptoms and signs in the textual descriptions of the EHR is a bit vague. The authors are describing that a new model was trained using a set of training and test notes and that a specific F1 was obtained. It is true the authors are claiming that their work is similar to a previous one where they created a symptom extractor for
--

	COVID-19. However, much more detail is needed in this context. Since the approach of this paper is focused on the identification of symptoms, and I suppose that a huge and relevant amount of the symptoms could be identified in the EHR, the part referred to as the NLP needs much more detail. Specifically, I think that the authors should provide as many technical details of how the models were trained and implemented. With this, I refer to two main points: 1) including concise descriptions of the clinical notes and how have been processed for the creation of the models, and 2) specific technical details of how the models were created. What frameworks were used, configuration details, etc.
--	---

VERSION 1 – AUTHOR RESPONSE

Reviewer Reports: Reviewer: 1 Dr. Ruth Swann, Cancer Research UK, NHS Digital

Comments to the Author: Thank you for the opportunity to review this paper. The authors address an important question and explain and present this well. I have made some comments as shown below.

1. *Strengths and limitations of the study* - As highlighted in the discussion, an additional limitation of this study is that the controls were not linked to SEER.
 - a. **Response:** We have added this limitation to the ‘Strengths and limitations’ section.
2. *Introduction* - ‘Screening for lung cancer remains low in the US.’ For the non-US audience, it would be useful to know the approximate uptake of the lung cancer screening programme in the US.
 - a. **Response:** We have added the lung cancer screening rate to the introduction.
3. *Methods* - As the data are collected from activity at UWM, a large tertiary care health centre, is it possible that cases or controls could have had consultations related to lung cancer in primary care or other hospitals before they were admitted to UWM?
 - a. **Response:** Yes, this is possible, but we tried to identify patients who had an established relationship with a UWM ambulatory care settings specifically in the 2 years before their first recorded lung cancer ICD code in the EHR.
4. *Methods* - Would the SEER diagnosis date be more accurate for the date of lung cancer diagnosis rather than the EHR diagnosis date?
 - a. **Response:** We discussed this at length and ultimately decided to use EHR diagnosis date because one of our co-authors who has had experience using SEER data for research advised us that it is difficult to rely on the SEER date of lung cancer diagnosis. The SEER date of diagnosis is based on the histological/pathological sampling date (e.g lung biopsy), whereas clinically a diagnosis of lung cancer may be made based on imaging findings (e.g. CT chest) prior to this.
5. *Methods* - To explain why the 3 month window was chosen, the following text in the discussion would be more useful in the methods ‘(to avoid potential seasonal differences in respiratory symptoms)’.
 - a. **Response:** We have moved this text to the methods section.
6. *Methods* - Could the authors briefly explain why cases with lymphoma, leukaemia, Kaposi sarcoma were excluded?
 - a. **Response:** Lymphoma, leukaemia, Kaposi sarcoma were excluded per the SEER ICD-O-3 Guidelines.
7. *Methods* - The methods describe that cases were excluded if they had evidence of previous tracheal cancer, mesothelioma, Kaposi sarcoma, lymphoma, or leukemia. Was evidence of these conditions searched for in the controls?
 - a. **Response:** We were not able to search for these in controls as these histology ICD codes were not easily identified in the EHR. We relied on SEER data to exclude cases of tracheal cancer, mesothelioma, Kaposi sarcoma, lymphoma, or leukemia, and we could not link controls to SEER data. We note this limitation in the Discussion section.

8. *Methods* - Why was race not included when matching controls to cases?
 - a. **Response:** The majority of patients in the UW system are white and finding rare populations to match did not seem an efficient use of our data resources. In addition, there is no particular biologic reason why race *per se* would be important to match on, rather than say smoking (or perhaps income or educational level) for which race may or may not be a proxy.
9. *Methods* - Was the analysis multivariate or multivariable? Both terms have been used in the paper.
 - a. **Response:** We have used a multivariable analysis. All mentions of multivariate have been changed to multivariable in the paper.
10. *Discussion* - Regarding the text 'Additionally, because we did not link to SEER for the control population, we were unable to apply two of the case exclusion criteria to our control sample: no current or prior history of lung cancer in SEER, although we did check the UW EHR for concurrent lung-cancer related ICD codes and medical history so this should be rare, and no prior history of tracheal cancer, mesothelioma, Kaposi sarcoma, lymphoma, or leukemia in SEER.' This is slightly confusing as the first part of the paragraph says that controls were not linked to SEER, then the end of the paragraph says that the controls had no prior history of the listed cancer types in SEER.
 - a. **Response:** We agree that this sentence is a bit confusing and have made edits to make it clearer in the Discussion section:
 'Additionally, because we did not link to SEER for the control population, we were unable to apply two of the case exclusion criteria to our control sample:
 1) no current or prior history of lung cancer in SEER, although we did check the UW EHR for concurrent lung-cancer related ICD codes and medical history so this should be rare, and 2) no prior history of tracheal cancer, mesothelioma, Kaposi sarcoma, lymphoma, or leukemia in SEER.'
11. *Discussion* - Could it be possible that a control had a lung cancer diagnosis after the index date? As some of these patients also have lung cancer related symptoms, could it be possible that they are just on an earlier part of the lung cancer diagnosis pathway?
 - a. **Response:** No, this is not possible (with the exception of missed cases by our clinicians.) For the entire study period, we matched ALL cases of lung cancer from SEER. If a control developed a diagnosis of lung cancer during our study period, they would have matched with SEER. It is possible that a control had a diagnosis several years after the index date and after our study. Therefore, while there is a risk of controls being misclassified as cases, this risk is small, particularly given the ratio of matching of controls to cases.
12. *Table 1* - Why are the Hispanic or Latino group displayed separately in the ethnicity category? There is no reference to this in the text.
 - a. **Response:** It is usual practice in US studies to refer to Hispanic or Latino as a separate ethnic category, however we have added a brief comment in the text about this finding.
13. *Figure 1* - Here it mentions that patients with first primary tumour located in anatomy other than lung are excluded, in the methods it mentions that cases were excluded if they had evidence of a history of any of the following cancers identified using histology codes in SEER: tracheal cancer, mesothelioma, Kaposi sarcoma, lymphoma, or leukemia (which I presume is histology code does not meet inclusion criteria in figure 1). Could you please confirm whether other primary tumours were excluded from the cases and if so include this in the methods.
 - a. **Response:** Thank you for clarifying, yes the Figure is correct and the Methods section was missing a critical piece of information in that we were only interested in *primary lung tumors not metastases to the lung* from another body site, we have modified the Methods section as follows:
 - i. 'Cases were excluded if they did not match with the SEER registry, or if they had a first primary tumor located in anatomy other than the lung, or had evidence of a history of any of the following cancers identified using histology

codes in SEER: tracheal cancer, mesothelioma, Kaposi sarcoma, lymphoma, or leukemia’.

14. *Figure 2* - As not all of the results are shown in the graph, I would suggest to add in the footnote to see Appendix 4 for complete set of results.
 - a. **Response:** We have added a footnote to Figure 2 indicating complete set of results can be found in Appendix 4.

Reviewer: 2 Dr. Alejandro Rodríguez-González , Universidad Politecnica de Madrid

Comments to the Author:

The paper is quite interesting and well-written. The methods are, in general, well described, as well as the results obtained.

1. My main concern is with the part where the NLP approach is described. The description of the NLP approach for the identification of potential symptoms and signs in the textual descriptions of the EHR is a bit vague. The authors are describing that a new model was trained using a set of training and test notes and that a specific F1 was obtained. It is true the authors are claiming that their work is similar to a previous one where they created a symptom extractor for COVID-19. However, much more detail is needed in this context. Since the approach of this paper is focused on the identification of symptoms, and I suppose that a huge and relevant amount of the symptoms could be identified in the EHR, the part referred to as the NLP needs much more detail. Specifically, I think that the authors should provide as many technical details of how the models were trained and implemented. With this, I refer to two main points: 1) including concise descriptions of the clinical notes and how have been processed for the creation of the models, and 2) specific technical details of how the models were created. What frameworks were used, configuration details, etc.
 - a. **Response:** Thank you for giving us the opportunity to add more detail to the Methods section. Indeed, as the Reviewer notes this is one of the novel parts of this study, and additional details of NLP are warranted. We have therefore added the following to the Methods section, and include a figure in the Appendices (Appendix 3) which provides additional technical explanation:
 - i. ‘Symptoms and signs were automatically extracted from free-text clinical notes using natural language processing (NLP), including notes for all visit types in the 2-year period. In previous work, we developed a deep learning symptom extraction model that generates structured semantic representations of symptoms.¹⁹ The annotation scheme and extraction architecture from this prior work represents symptoms using event-based approach. Each symptom event includes a trigger span that identifies the specific symptom (e.g. “cough” or “shortness of breath”) and multiple attributes that characterize the symptom. The attributes most relevant to this work are the *Assertion* value, which indicates whether the symptom is *present, absent, possible, etc.*, and the *Anatomy*, which indicates the anatomical location of the symptom (e.g. “chest wall” or “lower back”). Structured symptom predictions were generated using the Span-based Event Extractor architecture in Appendix 3. Each clinical note is split into sentences, which feed into the extractor. The words (tokens) of each sentence are mapped to a vector space using a clinical version of the Bidirectional Encoder Representations from Transformers (BERT) model (no model fine-tuning)²⁸,

²⁹. The BERT mapping of each sentence then feeds into a bidirectional Long Short-Term Memory (LSTM) network, which adapts the BERT encoding to the target extraction task. All possible token spans for the sentence are enumerated, and self-attention is used to create a representation for each span, $g_{c,i}$. Each of the enumerated spans is then classified using feedforward neural networks, ϕ_c , that operate on the span representation, $g_{c,i}$. The span scoring layer, ϕ_c , identifies the symptom triggers and attributes. Clinical notes frequently describe multiple symptoms within a sentence, and the relationships between the identified symptoms and attributes must be resolved. The identified symptom triggers are paired with the associated symptom attributes through the role scoring layer, ψ_d , which consists of a feedforward neural network that operates on span representation pairs. The output of the Span-based Event Extractor is a structured symptom representation, where identified symptoms are assigned multiple attributes.'

VERSION 2 – REVIEW

REVIEWER	Ruth Swann Cancer Research UK
REVIEW RETURNED	01-Feb-2023

GENERAL COMMENTS	Thank you for addressing all of the comments
--

REVIEWER	Alejandro Rodríguez-González Universidad Politecnica de Madrid
REVIEW RETURNED	22-Jan-2023

GENERAL COMMENTS	The authors have replied to my comment and most of my doubts have been addressed in a satisfactory way. Just two minor comments and one major:  - Please introduce a description of the type of clinical note(s) that were processed (consultation, admission, discharge, etc..). If possible, will be useful to include a screenshot of one example note (anonymizing relevant data). It is important also to add information about the number of notes that were processed, the number of patients processed in the NLP process, the average number of notes per patient, etc. - Please introduce more info about the used technological stack (libraries, programming language, etc). As major comment: Are the data (clinical notes) and/or the code/models available? trying to address as much reproducibility and transparency in research, it should be important to make this available, if possible. If not, please specify in the manuscript what is available and what it is not, and the reasons.
--

VERSION 2 – AUTHOR RESPONSE

Reviewer: 1 Dr. Ruth Swann, Cancer Research UK, NHS Digital

Comments to the Author: Thank you for addressing all of the comments

Author response *Noted*

Reviewer 2: Dr. Alejandro Rodríguez-González , Universidad Politecnica de Madrid

The authors have replied to my comment and most of my doubts have been addressed in a satisfactory way. Just two minor comments and one major:

- Please introduce a description of the type of clinical note(s) that were processed (consultation, admission, discharge, etc..). If possible, will be useful to include a screenshot of one example note (anonymizing relevant data). It is important also to add information about the number of notes that were processed, the number of patients processed in the NLP process, the average number of notes per patient, etc.

- Please introduce more info about the used technological stack (libraries, programming language, etc).

As major comment: Are the data (clinical notes) and/or the code/models available? trying to address as much reproducibility and transparency in research, it should be important to make this available, if possible. If not, please specify in the manuscript what is available and what it is not, and the reasons.

Author response

Page 7 provides details of the types of clinical notes that were processed for NLP. We do not feel it is necessary to include a screenshot of one example note as an example of how the NLP tool was applied to free text clinical notes, as we used multiple different types of notes (progress notes, hospital discharge notes, etc), and further do not feel that an anonymized note would add much insight to this, nor be particularly useful for the reader.

We have provided further information about the technological stack used on page 9, and two new citations (#22, 23) added also. We are able to release the symptom extractor, but are not permitted to release the data. The symptom extractor will be released through UW-BioNLP github <https://github.com/uw-bionlp> which is now noted on page 9 of the manuscript. Due to confidentiality requirements, we are not permitted to release the notes. Data release requires automatic de-identification and manual check of any potential auto de-id problems, which does not scale for such large datasets as used here